# CROWN: A Novel Approach to Comprehending Users' Preferences for Accurate Personalized News Recommendation

## ABSTRACT

Personalized news recommendation aims to assist users in finding news articles that align with their interests, which plays a pivotal role in mitigating users' information overload problem. Despite the breakthrough in personalized news recommendation, the following challenges have been rarely explored: (**C1**) *Comprehending manifold intents coupled within a news article*, (**C2**) *Differentiating varying post-read preferences of news articles*, and (**C3**) *Addressing the cold-start user problem*. To tackle these challenges together, we propose a novel personalized news recommendation framework (**CROWN**) that employs (1) category-guided intent disentanglement for (C1), (2) consistency-based news representation for (C2), and (3) GNN-enhanced hybrid user representation for (C3). Furthermore, we incorporate a category prediction into the training process of CROWN as an auxiliary task for enhancing intent disentanglement. Extensive experiments on two real-world datasets reveal that (1) CROWN outperforms twelve state-of-the-art news recommendation methods and (2) the proposed strategies significantly improve the accuracy of CROWN.

## KEYWORDS

Personalized news recommendation, news representation, user modeling, cold-start user problem

**ACM Reference Format:**
Anonymous Author(s). 2025. CROWN: A Novel Approach to Comprehending Users' Preferences for Accurate Personalized News Recommendation. In *Proceedings of the ACM Web Conference 2025 (WWW'25)*. ACM, New York, NY, USA, 12 pages. https://doi.org/10.1145/xxxxxxx.xx

## 1 INTRODUCTION

Personalized news recommendation aims to provide users with news articles that match their interests, playing a pivotal role in web-based news platforms such as Google News and MSN News in alleviating the information overload of users. For accurate personalized news recommendations, it is crucial to accurately model users' interests based on their historical clicked news [1, 2, 16, 24, 33, 36–38]. A general approach to modeling a user's interest, widely adopted in existing methods [3, 10, 14, 21, 23, 25, 29, 34, 35, 45], is as follows: (1) (**news representation**) the embeddings of news articles in a user's click history are generated by a news encoder and (2) (**user representation**) these news embeddings are then aggregated by a user encoder, thereby generating the user embedding.

**Challenges**. Although many existing methods have been proposed for better news and user representations, the following challenges still remain under-explored (See Figure 1):

*WWW'25, April 28–May 2, 2025, Sydney, Australia*

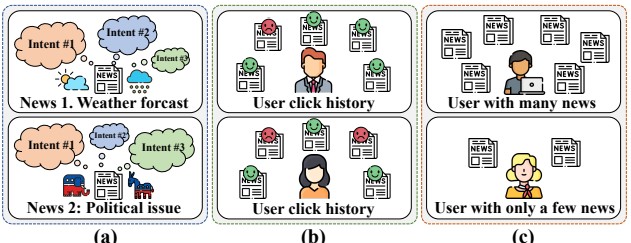

**Figure 1: Challenges of personalized news recommendation: (a) news articles are usually created with various intents, (b) users have varying post-read preferences to news articles, and (c) some users have only a few clicked news.**

(**C1**) **Comprehension of manifold intents**. Naturally, a news article can be created with a range of intents; they may differ across articles. For example, a weather forecast news aims to achieve multiple intents: (intent #1) to provide weather information and (intent #2) to improve public interest, but does not (intent #3) to persuade people. On the other hand, a political news may aim (intent #1) to provide information about a new policy and (intent #3) to persuade people to (dis)agree with the policy, but may not be interested in (intent #2) public interest. Hence, *it is critical to precisely comprehend each of intents coupled with a news article*.

(**C2**) **Differentiation of varying post-read preferences**. The news consumption process is as follows: a user clicks a news article if its *title* aligns with the user's interest; (Case #1) if the news *content* also aligns with her interest, she is happy to read it (i.e., high post-read preference); (Case #2) Otherwise, she is disappointed and closes it quickly (i.e., low post-read preference). In both cases, however, news articles are included in her click history; the articles that she is unlikely to prefer after clicking can cause inaccurate user modeling. Therefore, *it is necessary to differentiate news articles with different post-read preferences*.

(**C3**) **Cold-start user problem**. The *cold-start user problem* refers to the challenge of providing personalized recommendations to *new users* for whom the recommender system has little information. Since most existing news recommender systems rely on users' historical clicked news [10, 16, 20, 25, 33, 34, 36], it is challenging to capture new users' interests and make reliable recommendations to them. Thus, *it is essential to capture cold-start users' interests by using a small amount of their historical information*.

To address the aforementioned challenges, in this paper, we propose a novel framework for personalized news recommendation, named **CROWN**, which stands for **C**atego**R**y-guided intent disentanglement and c**O**nsistency-based ne**W**s represe**N**tation. CROWN consists of four modules: two encoding modules (news and user encoders) and two prediction modules (click and category predictors).

**News encoder**. The news encoder employs two key strategies:

(1) **Category-guided intent disentanglement**. We revisit the effect of a news category, which serves as beneficial metadata to

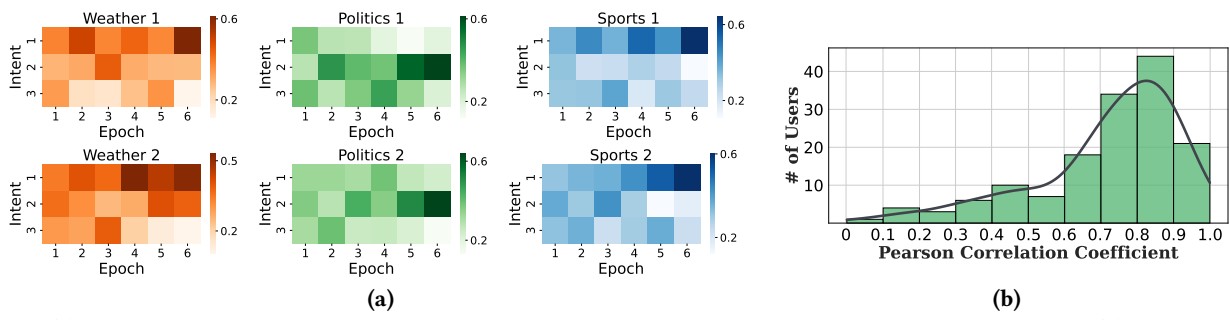

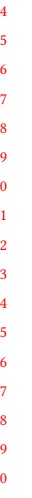

**Figure 2: (a)** Different intent distributions of news articles according to their categories in MIND [40] and **(b)** Distribution of Pearson correlation coefficients between the title-content consistency and users' content reading time in Adressa [5].

enhancing news representation [16, 21, 25, 29, 33, 35]. It is likely that the news articles within the *same category* have *similar intents* despite their different contents. As such, we posit that the news category can provide us with useful insights for better understanding various coupled intents of a news article. From this hypothesis, we propose *category-guided intent disentanglement* that represents a news article as multiple disentangled embeddings, each associated with a distinct intent, with the aid of its category information. To show the relation between the category and intents of a news article, we (1) randomly selected two news articles from each of three categories (Weather, Politics, and Sports), (2) represented each article into $k(=3)$ disentangled intent embeddings, and (3) computed the relative importance score of each intent embedding (See Appendix A.4 for more details). Figure 2(a) shows that news articles within the same category tend to show similar intent distributions even if their news contents differ as we claimed, whereas news articles from different categories exhibit different intent distributions.

**(2) Consistency-based news representation**. When a user clicks the *title* of a news article that aligns with her interest, she may expect that its *content* would align with her interest as well. Then, as its content matches her interest more, her *post-read preference* tends to be higher [17, 18, 43]. To this intuition, we hypothesize that *a user's post-read preference to her clicked news is correlated to the consistency between its title and content*. To verify our hypothesis, we conduct an experiment to compute the Pearson correlation coefficient between the title-content consistency and a user's content reading time (i.e., post-read preference). Figure 2(b) shows that *users' post-read preferences to their clicked news are closely related to the title-content consistency*. Motivated by this observation, we propose a method of *consistency-based news representation* that represents the news embedding by aggregating the title and content embeddings based on the degree of title-content consistency. Here, the title-content consistency plays a crucial role in adjusting how much the content is integrated into the news embedding, thereby differentiating varying post-read preferences for (C2).

**User encoder**. Then, based on the represented news embeddings, the user encoder generates a user embedding:

**(3) GNN-enhanced hybrid user representation**. In recommender systems, users with similar interests are likely to prefer similar items (i.e., news articles in our case) and the information about the users with similar interests (i.e., collaborative signals) tends to be beneficial in enhancing user representation [15, 21, 37], especially for the users with only a few historical clicked news

(i.e. the cold-start users). Based on this intuition, we adopt a *hybrid approach* to user representation that incorporates a graph neural network (GNN) into the user encoder. Therefore, the user encoder of CROWN not only aggregates the embeddings of a user's clicked news articles (i.e., content) but also leverages mutual complementary information from other users (i.e., collaborative signals), thereby alleviating the cold-start user problem for (C3).

**Click and category predictors**. The embeddings of a target user and a candidate article are fed into a click predictor to decide whether the user will click the candidate news (i.e., *primary task*). All of the model parameters of CROWN are learned based on the *click prediction loss*. In addition to the primary task, we incorporate a *category prediction* into the process of CROWN as an *auxiliary task*, which provides supplementary supervisory signals to guide the news encoder to be trained for better intent disentanglement.

**Contributions**. The main contributions of this work are as follows.

- **Challenges**: We identify three crucial yet under-explored challenges for accurate personalized news recommendation: (C1) comprehending manifold intents coupled within a news article, (C2) differentiating varying post-read preferences of news articles, and (C3) addressing the cold-start user problem.
- **Framework**: We propose a novel framework for personalized news recommendation, CROWN, that effectively addresses the three challenges by employing (1) category-guided intent disentanglement, (2) consistency-based news representation, and (3) GNN-enhanced hybrid user representation.
- **Evaluation**: We conduct experiments on real-world datasets to demonstrate that (1) (*accuracy*) CROWN consistently achieves the news recommendation accuracy higher than *all* state-of-the-art methods and (2) (*effectiveness*) each of the proposed strategies is significantly effective in improving the accuracy of CROWN.

For reproducibility, we have released the code of CROWN and the datasets at https://anonymous.4open.science/r/CROWN-7B75.

## 2 RELATED WORK

In this section, we review existing news recommendation methods. Table 1 compares CROWN with existing methods in terms of the three challenges. Many existing works [1, 10, 13, 20, 21, 25, 31, 33–36, 41] focus on modeling users' interests based on their clicked news articles with the news title, content, and category. For instance, NRMS [36] adopts self-attention networks to capture the context of news articles. NAML [33] employs a multi-view attention mechanism to learn various types of news information (e.g., title and

**Table 1: Comparison of existing methods with CROWN based on the challenges that we address.**

| Method | (C1) | (C2) | (C3) |
|---|---|---|---|
| LibFM [26], DSSM [9], NPA [34], NRMS [36], LSTUR [1] | - | - | - |
| NAML [33], TANR [35], FIM [29], HieRec [25], MINER [13], MCCM [31] | ✓ | - | - |
| CNE-SUE [21], DIGAT [20] | ✓ | - | ✓ |
| CAST [10] | - | ✓ | - |
| GLORY [41] | - | - | ✓ |
| CPRS [38], FeedRec [39], TCAR [4] | - | ✓ | - |
| HyperNews [16] | ✓ | ✓ | - |
| **CROWN** (proposed) | ✓ | ✓ | ✓ |

content). TANR [35] exploits topic information within news articles for news representation. NPA [34] employs personalized attention networks to selectively learn important information according to individual user preferences. LSTUR [1] considers both long-term and short-term user interests for better user representation. CNE-SUE [21] adopts a collaborative news encoder for mutual learning of title and content embeddings, along with a GCN-based user encoder. HieRec [25] utilizes a hierarchical user modeling method to capture users' diverse and multi-grained interests. CAST [10] leverages the news body text to enhance the words in a news title by using context-aware attention networks. MINER [13] employs a method to disentangle a user's interest into multi levels for better user representation. DIGAT [20] adopts a dual-graph (news-graph and user-graph) interaction process to match news and user representations. GLORY [41] enhances personalized recommendation by combining global user representations with local user representations. MCCM [31] reduces the negative impact of the noisy data via channel-wise dynamic news encoder and contrastive user encoder.

Historical clicked news articles alone, however, often struggle with achieving desirable performance [17, 18, 38, 39]. Thus, there have been a handful of methods that leverage additional user feedback [4, 16, 37–39]. For example, HyperNews [16] considers the freshness of news articles and users' active time. CPRS [38] takes users' reading satisfaction based on their reading speed into account in user representation. FeedRec [39] considers various user feedback types including 'skip,' 'finish,' 'share,' and 'quick close' to capture users' interests comprehensively. TCAR [4], a session-based news recommendation method, incorporates users' positive, negative, and neutral implicit feedback in user representation.

In real-world scenarios, however, such types of additional user feedback are not always available [21, 25, 35, 37]. As such, it is important and practical to learn news and user representations solely based on users' clicked news articles and their static information, without relying on additional user feedback. To our best knowledge, this is the first work to address the three challenges without any additional user feedback.

## 3  PROBLEM DEFINITION

In this work, we consider the two following problems: click prediction (*primary* task) and category prediction (*auxiliary* task). The notations used in this paper are summarized in Table 2.

**Table 2: The notations and their descriptions.**

| Notation | Description |
|---|---|
| $U, N$ | a set of users and news articles |
| $T_n$ | a set of the word embeddings in the news title |
| $C_n$ | a set of the word embeddings in the news content |
| $c_n, sc_n$ | the news category and sub-category |
| $\mathbf{r}^u, \mathbf{r}^n$ | the user/news representations |
| $\mathbf{r}^n_{(T)}, \mathbf{r}^n_{(C)}$ | the news title/content representations |
| $\mathbf{r}^n_{(T,k)}, \mathbf{r}^n_{(C,k)}$ | $k$-th intent embeddings of the news title/content |
| $d$ | the embedding dimensionality |
| $K$ | the number of intents |
| $\Theta_U, \Theta_N$ | user and news encoders |
| $\Phi_P, \Phi_A$ | click and category predictors |
| $\mathbf{W}, \mathbf{b}$ | trainable weights and biases |
| $\mathcal{L}(\cdot)$ | loss function |
| $\eta$ | user-defined learning rate |

**PROBLEM 1 (CLICK PREDICTION).** *Given a target user $u_t$ and a candidate news article $n_c$, the goal is to predict whether the target user $u_t$ will click the candidate news $n_c$ or not.*

To solve this problem, we first generate the representation of the target user $u_t$ based on her clicked news articles $N_{u_t} = \{n^{u_t}_1, n^{u_t}_2, \ldots, n^{u_t}_{|N_{u_t}|}\}$. Each news article is denoted by $n = (T_n, C_n, c_n, sc_n)$, where $T_n \in \mathbb{R}^{|T_n| \times d}$ is a set of the word embeddings in the news title, i.e., $\{w^n_1, w^n_2, \ldots, w^n_{|T_n|}\}$, $C_n \in \mathbb{R}^{|C_n| \times d}$ is a set of the word embeddings in the content, i.e., $\{w^n_1, w^n_2, \ldots, w^n_{|C_n|}\}$, $c_n$ is the news category, and $sc_n$ is the news sub-category. Specifically, a news encoder $\Theta_N$ generates the representation of each clicked news article $n \in N_{u_t}$, i.e., $\Theta_N(n) \rightarrow \mathbf{r}^n$; then, a user encoder $\Theta_U$ generates the target user representation by aggregating the news representations, i.e., $\Theta_U(\mathbf{R}^{u_t}) \rightarrow \mathbf{r}^{u_t}$, where $\mathbf{R}^{u_t} = \{\mathbf{r}^n | n \in N_{u_t}\}$. Finally, the representations of the target user and candidate news are fed into a click predictor $\Phi_P$ to decide whether $u_t$ will click $n_c$ or not, i.e., $\Phi_P(\mathbf{r}^{u_t}, \mathbf{r}^{n_c}) \rightarrow \hat{y} \in \{0, 1\}$.

In addition to the click prediction (i.e., primary task), we integrate category prediction in the training process of CROWN as an *auxiliary task*, providing complementary information to better train the news encoder $\Theta_N$.

**PROBLEM 2 (CATEGORY PREDICTION).** *Given a news article $n$, the goal is to predict to which category the news $n$ belongs.*

For each candidate news $n_c$, its news representation is generated by a news encoder $\Theta_N$ and fed into a category predictor $\Phi_A$ to decide which category the candidate article $n_c$ belongs to, i.e., $\Phi_A(\mathbf{r}^{n_c}) \rightarrow \hat{z} \in \{1, 2, \ldots, |c|\}$, where $|c|$ is the number of categories.

Based on these two tasks, all model parameters of CROWN are jointly optimized to minimize both click and category prediction losses in *an end-to-end way*. We will describe each module of CROWN and loss function $\mathcal{L}(\cdot)$ in Section 4.

## 4  CROWN: PROPOSED FRAMEWORK

In this section, we present a novel personalized news recommendation framework, named **C**atego**R**y-guided intent disentanglement and c**O**nsistency-based ne**W**s represe**N**tation (**CROWN**).

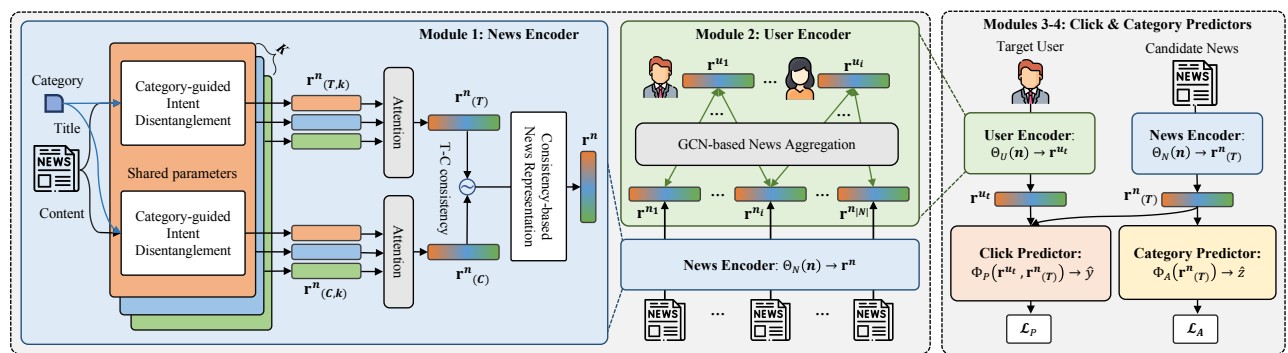

**Figure 3: Overview of CROWN: two encoding modules (Modules 1-2) and two prediction modules (Modules 3-4).**

**Overview**. As illustrated in Figure 3, CROWN consists of four modules: two encoding modules (i.e., news and user encoders) and two prediction modules (i.e., click and category predictors). Given a target user and a candidate news, CROWN proceeds as follows:

- **News encoder** (Module #1) generates the representations of the news articles in the target user's click history;
- **User encoder** (Module #2) aggregates these news representations into the target user representation;
- **Click predictor** (Module #3) predicts whether the target user will click the candidate news based on their representations;
- **Category predictor** (Module #4) predicts which category the candidate news belongs to.

## 4.1 Module 1: News Encoder

The news encoder of CROWN employs (1) category-guided intent disentanglement and (2) consistency-based news representation to address the two challenges: (C1) comprehending a range of intents coupled within a news article and (C2) differentiating news articles with varying post-read preferences.

**(1) Category-guided intent disentanglement.** As explained in Section 1, the news category, which serves as beneficial metadata to enhancing news representation, has been widely used in many existing news recommendation methods [16, 21, 25, 29, 33, 35]. Intuitively, it is likely that the news articles belonging to the *same* category might have *similar* intents, even if they provide different news content. For example, two weather news articles, while conveying different weather information, are likely to have similar intents of providing weather information (intent #1) and increasing public interest (intent #2). Similarly, two political news articles, despite different political stances, may aim to provide information about the same policy (intent #1), with the intent to persuade people to (dis)agree with a specific viewpoint (intent #3).

Based on this intuition, we posit that a range of intents coupled within a news article are closely related to its category. From this hypothesis, we propose a method of *category-guided intent disentanglement* that represents each of the title and content of a given news article as *multiple disentangled embeddings*, each associated with a distinct intent, with the aid of its category information.

Specifically, given a news article $n = (T_n, C_n, c_n, sc_n)$ in a user's click history and the number of intents $K$, CROWN represents the news title $T_n$ and content $C_n$ as $K$ intent embeddings, i.e., $\mathbf{r}^n_{(T,k)}$ and $\mathbf{r}^n_{(C,k)}$, where each embedding corresponding to $k$-th intent.

- **(1)-(a) Text embedding**: We apply the multi-head attention block (MAB) [12, 27] to $T_n = \{w_1, w_2, ..., w_{|T_n|}\}$ and $C_n = \{w_1, w_2, ..., w_{|C_n|}\}$ to generate *context-aware* title and content embeddings, i.e., $\mathbf{r}^n_T \in \mathbb{R}^d$ and $\mathbf{r}^n_C \in \mathbb{R}^d$, while considering the different informativeness of each word in the news title and content, where $w_i$ is the pre-trained word embeddings by Glove [22]:

$$\mathbf{r}^n_T = Avg(\text{MAB}(T_n, T_n)), \quad \mathbf{r}^n_C = Avg(\text{MAB}(C_n, C_n)), \quad (1)$$

where $\text{MAB}(X, Y) = \text{LayerNorm}(H + \text{FeedForward}(H))$ and $H = \text{LayerNorm}(X + \text{Multihead}(X, Y, Y))$.

- **(1)-(b) Category embedding**: We then generate the category embedding, i.e., $\mathbf{c}^n_* \in \mathbb{R}^{d_c}$, by mixing the information of the news category and sub-category:

$$\mathbf{c}^n_* = Mix(c_n, sc_n) = (c_n \oplus sc_n) \cdot \mathbf{W}_{mix} + \mathbf{b}_{mix} \quad (2)$$

where $\oplus$ is the concatenation operator and, $\mathbf{W}_{mix}$ and $\mathbf{b}_{mix}$ are trainable parameters.

- **(1)-(c) Intent disentanglement**: We generate $K$ intent embeddings for each of the news title and content, i.e., $\mathbf{r}^n_{(T,k)} \in \mathbb{R}^d$ and $\mathbf{r}^n_{(C,k)} \in \mathbb{R}^d$, by disentangling multiple intents from the title and content embeddings:

$$\mathbf{r}^n_{(T,k)} = DisEntangle(\mathbf{r}^n_T, \mathbf{c}^n_*) = \sigma((\mathbf{r}^n_T \oplus \mathbf{c}^n_*) \cdot \mathbf{W}_k + \mathbf{b}_k), \quad (3)$$

$$\mathbf{r}^n_{(C,k)} = DisEntangle(\mathbf{r}^n_C, \mathbf{c}^n_*) = \sigma((\mathbf{r}^n_C \oplus \mathbf{c}^n_*) \cdot \mathbf{W_k} + \mathbf{b}_k) \quad (4)$$

where $\sigma(\cdot)$ is a non-linear activation function (ReLU) and, $\mathbf{W}_k$ and $\mathbf{b}_k$ are the trainable parameters of the $k$-th intent disentanglement layer. As highlighted by the same colors in Figure 3, the weights and biases are shared in disentangling the intents from the title and content embeddings.

Note that, although there have been a handful works that employ a method to learn disentangled representations for accurate recommendation [13, 19, 32, 44], they focus on *disentangling a user's interest* for better user representation, rather than *a news article's intents* for better news representation. In other words, these existing works are *orthogonal* to our work and could be incorporated into the user encoder of our method.

**(2) Consistency-based news representation.** We then generate the final news embedding by aggregating the disentangled title and content embeddings. As mentioned in Section 1, we observed that "*the title-content consistency of a news article is strongly correlated to users' post-read preferences to the new article*" via the preliminary experiment (See Figure 2). Based on this observation, we propose

a method of *consistency-based news representation* that aggregates the title and content embeddings into the final news embedding, based on the degree of title-content consistency.

- **(2)-(a) Intent-aware aggregation**: Given $K$ intent embeddings for each of the news title and content obtained in the previous step, i.e., $\mathbf{r}^n_{(T,k)}$ and $\mathbf{r}^n_{(C,k)}$, we aggregate the $K$ intent embeddings of each of the news title and content into a single intent-aware embedding, i.e., $\mathbf{r}^n_{(T)} \in \mathbb{R}^d$ and $\mathbf{r}^n_{(C)} \in \mathbb{R}^d$, by using an *attention mechanism* to give different significance for each intent:

$$\mathbf{r}^n_{(T)} = \sum_{k=1}^{K} \alpha_{(T,k)} \cdot \mathbf{r}^n_{(T,k)}, \quad \mathbf{r}^n_{(C)} = \sum_{k=1}^{K} \alpha_{(C,k)} \cdot \mathbf{r}^n_{(C,k)}, \quad (5)$$

where $\alpha_{(*,k)} = \frac{exp(z_{(*,k)})}{\sum_{i=1}^{K} exp(z_{(*,i)})}$ indicates the attention weight of the $k$-th intent embedding; $z_{(*,k)} = \tanh(\mathbf{W}_{agg} \cdot \mathbf{r}^n_{(*,k)} + \mathbf{b}_{agg})$; $\mathbf{W}_{agg}$ and $\mathbf{b}_{agg}$ are trainable parameters. Thus, $\mathbf{r}^n_{(*)}$ is the '*intent-aware*' title/content embedding that understands a range of intents coupled within a news article.

- **(2)-(b) Consistency computation**: Then, we compute the consistency score between the news title and content, i.e., $cs^n$, based on the intent-aware title and content embeddings $\mathbf{r}^n_{(T)}$ and $\mathbf{r}^n_{(C)}$:

$$cs^n = Consistency(\mathbf{r}^n_{(T)}, \mathbf{r}^n_{(C)}) = \frac{Cos(\mathbf{r}^n_{(T)}, \mathbf{r}^n_{(C)}) + 1}{2}. \quad (6)$$

As a consistency measure, we use the cosine similarity and rescale the result to the range of [0,1].

- **(2)-(c) News representation**: Finally, we generate the final news embedding, i.e., $\mathbf{r}^n \in \mathbb{R}^d$, by aggregating its intent-aware title and content embeddings $\mathbf{r}^n_{(T)}$ and $\mathbf{r}^n_{(C)}$ based on its consistency $cs^n$:

$$\mathbf{r}^n = \mathbf{r}^n_{(T)} + cs^n \cdot \mathbf{r}^n_{(C)}. \quad (7)$$

Therefore, the news *content* information is *differentially incorporated* into news representation, guided by the title-content consistency, playing a crucial role in adjusting how much the news content information is integrated into the final news embedding.

As a result, the news encoder of CROWN is able to effectively address the two challenges of news representation by employing the (1) category-guided intent disentanglement for (C1) and (2) consistency-based news representation for (C2). We will verify the effectiveness of the proposed strategies in improving the recommendation accuracy of CROWN in Section 5.3 and Appendix A.4.

## 4.2 Module 2: User Encoder

Many existing news recommendation methods suffer from (C3) the cold-start user problem since they rely only on users' historical clicked news [10, 16, 20, 25, 33, 34, 36]. To alleviate this challenge, we design (3) a GNN-enhanced hybrid user encoder that not only aggregates the embeddings of a user's clicked news articles but also leverages mutual complementary information *from other users* (i.e., *collaborative signals*) via a graph neural network (GNN).

(3) **GNN-enhanced hybrid user representation.** Given a user $u$ and its clicked news articles $N_u = \{n^u_1, ..., n^u_{N_u}\}$, CROWN generates the user embedding, i.e., $\mathbf{r}^u$ by aggregating the embeddings of its clicked news articles, i.e., $\{\mathbf{r}^n | n \in N_u\}$.

- **(3)-(a) Graph construction**: First, we construct a user-news bipartite graph (See Module 2 in Figure 3), where each node corresponds to a user (upper) or a news article (lower) and an edge corresponds to a user's click. Note that only user-news edges exist in the bipartite graph (i.e., no user-user or news-news edges). We represent the bipartite graph as a matrix $\mathbf{A} \in \mathbb{R}^{|U| \times |N|}$. The node feature matrix for user nodes, i.e., $\mathbf{R}^u \in \mathbb{R}^{|U| \times d}$, are randomly initialized and that for news nodes, i.e., $\mathbf{R}^n \in \mathbb{R}^{|N| \times d}$ are initialized with the news embeddings represented by the news encoder.

- **(3)-(b) GNN-based update**: We apply a graph neural network (GNN) to the user-news bipartite graph to mutually update the user and news embeddings. Given a user-news matrix $\mathbf{A}$ and the node feature matrices $\mathbf{R}^u$ and $\mathbf{R}^n$, the user and news embeddings at the $l$-th layer are defined as follows:

$$\mathbf{R}^u = \sigma(\mathbf{A}\mathbf{R}^n\mathbf{W}^l_g + \mathbf{b}^l_g), \quad \mathbf{R}^n = \sigma(\mathbf{A}^\top\mathbf{R}^u\mathbf{W}^l_g + \mathbf{b}^l_g), \quad (8)$$

where $\mathbf{W}^l_g$ and $\mathbf{b}^l_g$ are trainable parameters. For simplicity, we omit the normalized terms. It is worth noting that any state-of-the-art GNN models [6–8, 11, 28] can be applied to the user encoder of CROWN since our method is *agnostic* to the user encoder architecture. We use GAT [28] as a GNN model in this work. The impacts of four state-of-the-art GNN models on the accuracy of CROWN are included in Appendix A.5.

- **(3)-(c) User representation**: Finally, we generate the final user embedding, i.e., $\mathbf{r}^u$. We aggregate all news embeddings in the user's click history based on their attention weights computed by the *target user-wise attention* mechanism, to reflect different importance of each news article in the context of the target user:

$$\mathbf{r}^u = \sum_{i=1}^{|N_{u_t}|} \alpha_i \cdot \mathbf{r}^n_i, \quad (9)$$

where $\alpha_i = \frac{exp(z_j)}{\sum_{j=1}^{|N_{u_t}|} exp(z_j)}$ indicates the attention weight of the $i$-th news embedding; $z_j = \mathbf{q}^\top \cdot \tanh(\mathbf{W}_{user} \cdot \mathbf{r}^n_j + \mathbf{b}_{user})$; $\mathbf{q}$, a query vector, is the target user embedding learned from the GNN model; and $\mathbf{W}_{user}$ and $\mathbf{b}_{user}$ are trainable parameters.

We will evaluate the effectiveness of our user encoder in improving the recommendation accuracy of CROWN in Section 5.4.

## 4.3 Modules 3-4: Click and Category Predictors

Based on the learned news and user representations, CROWN performs click prediction (primary) and category prediction (auxiliary), and then, updates the model parameters based on the two losses.

**Click Prediction.** Given a target user and a candidate news article, the click predictor computes the probability of the user clicking the candidate article based on their representations. Note that we use only the title embedding of a candidate news, i.e., $\mathbf{r}^n_{(T)}$ for click prediction since a user could see the news *title only*, not the news content, when deciding whether to click a news article or not. Following [1, 16, 21, 25, 33–36, 39], we compute the click probability by the inner product of the user embedding and the news title embedding, i.e., $\hat{y}_{u,n} = \mathbf{r}^u \cdot \mathbf{r}^n_{(T)}$.

We construct the click prediction loss with negative examples in the same way as in [10, 21, 23, 24, 30, 33, 36, 38]. Specifically, for

each news article clicked by a target user (i.e., positive example), we randomly sample $M$ news articles not clicked by the user as negative examples. We aim to optimize the model parameters of CROWN so that positive examples obtain higher scores than negative examples. Formally, the click prediction loss (primary loss) $\mathcal{L}_P$ is defined as:

$$\mathcal{L}_P = -\sum_{i \in S} \log \left( \frac{exp(\hat{y}_i^+)}{exp(\hat{y}_i^+) + \sum_{j=1}^{M} exp(\hat{y}_{i,j}^-)} \right), \quad (10)$$

where $S$ is the set of positive examples, $\hat{y}_i^+$ is the $i$-th positive example, and $\hat{y}_{i,j}^-$ is the $j$-th negative example for $\hat{y}_i^+$.

**Category Prediction.** In addition, we incorporate a category prediction into the training process as an *auxiliary task*, which provides supplementary supervisory signals to guide the news encoder to be trained for better intent disentanglement (i.e., category-guided).

We compute the category probability distribution of a candidate news article by passing its title embedding to the category prediction layer, which consists of a fully-connected layer ($d \times |c|$), followed by a softmax layer, i.e., $\hat{z} = softmax(FC(\mathbf{r}_{(T)}^n))$. Formally, the category prediction loss (auxiliary loss) $\mathcal{L}_A$ is defined as:

$$\mathcal{L}_A = -\frac{1}{|N|} \sum_{i=1}^{|N|} \sum_{j=1}^{|c|} z_{i,j} \cdot \log(\hat{z}_{i,j}), \quad (11)$$

where $|N|$ is the number of news articles, $|c|$ is the number of categories, and $z_{i,j}$ and $\hat{z}_{i,j}$ are the ground-truth and predicted probabilities of the $i$-th news article being in the $j$-th category, respectively. Finally, we unify the click and category prediction losses by a weighted sum. Thus, the total loss is defined as:

$$\mathcal{L} = \mathcal{L}_P + \beta \mathcal{L}_A, \quad (12)$$

where $\beta$ controls the weight of the auxiliary task. Accordingly, all model parameters are trained to jointly optimize the two tasks. We will validate the impact of the hyperparameter $\beta$ on the accuracy of CROWN in Section 5.5. We also analyze the time and space complexity of CROWN and provide the results in Appendix A.1.

## 5 EXPERIMENTAL VALIDATION

In this section, we comprehensively evaluate CROWN by answering the following evaluation questions:

- **EQ1** (*Accuracy*): To what extent does CROWN improve the accuracy of existing methods in personalized news recommendation?
- **EQ2** (*News representation*): Is each of our strategies in a news encoder effective for getting better news representation?
- **EQ3** (*User representation*): Is our strategy in a user encoder effective for getting better user representation?
- **EQ4** (*Auxiliary task*): Does our auxiliary task contribute to the improvement of CROWN in accuracy?
- **EQ5** (*Cold start user problem*): Is CROWN effective in addressing the cold start problem?

### 5.1 Experimental Setup

**Datasets and competitors.** We evaluate CROWN with two real-world datasets, **MIND-small**[1] (simply **MIND** hereafter) [40] and

---

[1]In our experiments, we used only **MIND-small** not **MIND-large** due to the limited computing power of our experimental environment, which may limit the generalizability of CROWN. However, it is worth noting that the trends in recommendation

**Table 3: Statistics of news article datasets.**

| Dataset | MIND-small [40] | Adressa-1week [5] |
|---|---|---|
| *# of users* | 94,057 | 601,215 |
| *# of news articles* | 65,238 | 73,844 |
| *# of clicks* | 347,727 | 2,107,312 |
| *# of categories* | 18 (270) | 108 (151) |
| *# of words-per-title* | 11.67 | 6.63 |
| *# of words-per-content* | 41.01 | 552.15 |

**Adressa-1week** (simply **Adressa** hereafter) [5]. Table 3 shows the statistics of the news article datasets. We compare CROWN with 12 state-of-the-art news recommendation methods: LibFM [26], DSSM [9], NPA [34], NRMS [36], NAML [33], LSTUR [1], FIM [29], HieRec [25], CNE-SUE [21], DIGAT [20][2], GLORY [41], and MCCM [31]. For LibFM, DSSM, FIM, HieRec, CNE-SUE, and DIGAT, we use the official source codes provided by the authors, for NPA, NAML, NRMS, and LSTUR, we use the implementations publicly available in the popular open-source library (MS open source), and for MCCM, we use their *best* reported results in [31] since its official source code is unavailable. Note that we use the same pre-trained word embeddings by GloVe [22] for all methods in order to fairly evaluate the recommendation.

**Evaluation protocol.** Following [16, 21, 24, 39], we evaluate the top-$n$ recommendation accuracy of each method. As evaluation metrics, we use area under the curve (AUC), mean reciprocal rank (MRR), and normalized discounted cumulative gain (nDCG). We measure AUC, MRR, nDCG@5, and nDCG@10 on the test set when the AUC on the validation set is maximized; we report the averaged AUC, MRR, nDCG@5, and nDCG@10 on the test set over five runs. The implementation details are provided in Appendix A.2.

### 5.2 EQ1. Model Accuracy

Table 4 shows the accuracies of all competing methods in two real-world datasets. The results demonstrate that CROWN *consistently* outperforms *all* state-of-the-art in *both* datasets in *all* metrics (averaged AUC, MRR, and nDCG@K). Specifically, CROWN achieves higher AUC, MRR, nDCG@5, and nDCG@10 by up to 4.78%, 11.71%, 18.82%, and 11.38%, respectively than CNE-SUE, the best competitor in **Adressa**. We note that these improvements of CROWN over the best competitor are remarkable, given that CNE-SUE has already improved other existing methods significantly [21]. In addition, we have conducted the $t$-tests with a 95% confidence level and verified that the improvement of CROWN over all competing methods are statistically significant (i.e., the $p$-values are below 0.05). Consequently, these results demonstrate that CROWN successfully addresses the three challenges of personalized news recommendation through the proposed strategies: (1) the category-guided intent disentanglement for (C1) understanding a range of intents coupled within a news article, (2) the consistency-based news representation for (C2) differentiating users' varying post-read preferences to a

---

accuracy of existing methods are consistent across the two datasets [13, 20]. Also, we used two news datasets from different sources (i.e., **MIND-small** and **Adressa-1week**) for comprehensive evaluation as in [2, 4, 41, 42].

[2]Following [41], we use DIGAT without the pre-trained language model (PLM) for fair comparison.

**Table 4: News recommendation accuracy on two real-world datasets: CROWN consistently outperforms *all* competing methods in terms of *all* metrics (The bold font and underline indicate the best and the second-best results, respectively).**

| | | LibFM | DSSM | NPA | NRMS | NAML | LSTUR | FIM | HieRec | CNE-SUE | DIGAT | GLORY | MCCM* | **CROWN** | Gain |
|---|---|---|---|---|---|---|---|---|---|---|---|---|---|---|---|
| **MIND** | AUC | 0.6039 | 0.6235 | 0.6542 | 0.6623 | 0.6648 | 0.6612 | 0.6571 | 0.6718 | 0.6761 | 0.6713 | 0.6792 | 0.6795 | **0.6857** | +0.91% |
| | MRR | 0.2835 | 0.2934 | 0.3092 | 0.3134 | 0.3142 | 0.3121 | 0.3093 | 0.3169 | 0.3219 | 0.3225 | 0.3221 | 0.3276 | **0.3405** | +3.94% |
| | nDCG@5 | 0.3040 | 0.3094 | 0.3408 | 0.3453 | 0.3480 | 0.3461 | 0.3428 | 0.3532 | 0.3572 | 0.3584 | 0.3601 | 0.3662 | **0.3783** | +3.30% |
| | nDCG@10 | 0.3650 | 0.3748 | 0.4039 | 0.4135 | 0.4112 | 0.4056 | 0.4047 | 0.4161 | 0.4203 | 0.4195 | 0.4196 | 0.4266 | **0.4384** | +2.77% |
| **Adressa** | AUC | 0.6105 | 0.6682 | 0.6704 | 0.7067 | 0.7612 | 0.7673 | 0.7324 | 0.7867 | 0.8074 | 0.7385 | 0.7638 | - | **0.8460** | +4.78% |
| | MRR | 0.3357 | 0.3657 | 0.3885 | 0.4168 | 0.4326 | 0.5034 | 0.4252 | 0.4922 | 0.5073 | 0.4361 | 0.4653 | - | **0.5667** | +11.71% |
| | nDCG@5 | 0.3083 | 0.3803 | 0.3955 | 0.4166 | 0.4453 | 0.4906 | 0.4318 | 0.4872 | 0.5021 | 0.4319 | 0.4761 | - | **0.5966** | +18.82% |
| | nDCG@10 | 0.3760 | 0.4163 | 0.4279 | 0.4725 | 0.5133 | 0.5577 | 0.4874 | 0.5667 | 0.5712 | 0.4958 | 0.5418 | - | **0.6362** | +11.38% |

*For MCCM [31], there is no reported accuracy on the **Adressa** dataset.

**Table 5: Ablation study of the news encoder: the proposed components of the news encoder significantly contribute to enhancing the model accuracy of CROWN (The underline indicate the best result when the same number of the components are applied).**

| | | Baseline | I.D. | C.I. | C.R. | I.D. C.I. | I.D. C.R. | C.I. C.R. | I.D. C.I. C.R. | Gain |
|---|---|---|---|---|---|---|---|---|---|---|
| **MIND** | AUC | 0.6624 | 0.6725 | 0.6731 | 0.6684 | 0.6764 | 0.6806 | 0.6793 | **0.6857** | +3.52% |
| | MRR | 0.3139 | 0.3201 | 0.3192 | 0.3182 | 0.3224 | 0.3282 | 0.3246 | **0.3405** | +8.47% |
| | nDCG@5 | 0.3484 | 0.3546 | 0.3558 | 0.3535 | 0.3567 | 0.3651 | 0.3611 | **0.3783** | +8.58% |
| | nDCG@10 | 0.4095 | 0.4171 | 0.4182 | 0.4151 | 0.4194 | 0.4259 | 0.4230 | **0.4384** | +7.06% |
| **Adressa** | AUC | 0.7789 | 0.8088 | 0.7977 | 0.7829 | 0.8215 | 0.8385 | 0.8343 | **0.8460** | +8.61% |
| | MRR | 0.4831 | 0.5094 | 0.5047 | 0.5006 | 0.5200 | 0.5284 | 0.5277 | **0.5667** | +17.30% |
| | nDCG@5 | 0.4904 | 0.5266 | 0.5199 | 0.4991 | 0.5331 | 0.5533 | 0.5518 | **0.5966** | +21.66% |
| | nDCG@10 | 0.5546 | 0.5886 | 0.5868 | 0.5704 | 0.5913 | 0.6035 | 0.6032 | **0.6362** | +14.71% |

news article, and (3) the GNN-enhanced hybrid user representation for (C3) the cold-start user problem.

## 5.3 EQ2. News Representation

We verify the effectiveness of *our proposed strategies in the news encoder*. We consider all possible combinations of the three components: **I.D.**: Intent disentanglement; **C.I.**: Category information; and **C.R.**: Consistency-based news representation.

Table 5 shows the results of the ablation study. Overall, each of our proposed strategies is always beneficial to improving the accuracy of CROWN. Specifically, when all strategies (**I.D. C.I. C.R.**) are applied to CROWN, the averaged AUC, MRR, nDCG@5, and nDCG@10 are improved by 8.61%, 17.30%, 21.66%, and 14.71%, respectively in **Adressa**, compared to the baseline version of CROWN. These results demonstrate that (1) the two challenges of news representation are critical for accurate news recommendation and (2) our proposed strategies of CROWN address them successfully.

Looking more closely, when one component is applied, CROWN with **I.D.** achieves the accuracy comparable to or higher than CROWN with **C.I.** and always outperforms CROWN with **C.R.**. This result implies that the precise comprehension of manifold intents of a news article is most important for accurate news recommendation, thereby verifying the effect of the *intent disentanglement* as we claimed in Section 4.1. When two components are applied, CROWN with **I.D. C.R.** consistently achieves the highest accuracy, compared with the other two versions. This is because CROWN with **I.D. C.R.** addresses the both challenges of news representation (**I.D.** for (C1) and **C.R.** for (C2)), while CROWN with **I.D. C.I.** or CROWN with **C.I. C.R.** tackles only one of the two challenges. Lastly, when all the components are applied, CROWN achieves the best results in *all* cases. This result demonstrates that the category information (**C.I.**) of a news article is able to provide useful insights for better intent disentanglement as we claimed in Section 4.1. In addition to the quantitative effect of the category

**Table 6: The model accuracy of CROWN according to different user encoders: the proposed GNN-enhanced hybrid user encoder consistently outperforms other user encoders.**

| | | AVG | ATT | GNN | **Hybrid** | Gain |
|---|---|---|---|---|---|---|
| **MIND** | AUC | 0.6720 | 0.6736 | 0.6764 | **0.6857** | +2.04% |
| | MRR | 0.3194 | 0.3236 | 0.3230 | **0.3405** | +6.61% |
| | nDCG@5 | 0.3550 | 0.3594 | 0.3600 | **0.3783** | +6.56% |
| | nDCG@10 | 0.4174 | 0.4208 | 0.4211 | **0.4384** | +5.03% |
| **Adressa** | AUC | 0.8216 | 0.8349 | 0.8257 | **0.8460** | +2.97% |
| | MRR | 0.5141 | 0.5223 | 0.5208 | **0.5667** | +10.23% |
| | nDCG@5 | 0.5295 | 0.5448 | 0.5316 | **0.5966** | +12.67% |
| | nDCG@10 | 0.5809 | 0.5851 | 0.5905 | **0.6362** | +9.52% |

information, we analyze the qualitative effect of the category information in disentangling multiple coupled intents and provide the results in Appendix A.4. We also evaluate the impact of the number of intents $K$ on the accuracy of CROWN and provide the results in Appendix A.3.

## 5.4 EQ3. User Representation

We verify the impact of the user encoder on the accuracy of CROWN. We compare four variants of CROWN with different user encoders. (1) AVG averages the embeddings of the news articles in a user's click history; (2) ATT aggregates the news embeddings based on their attention weights; (3) GNN mutually updates the news and user embeddings using a graph neural network (GNN); and (4) **Hybrid** is the proposed GNN-enhanced user encoder (i.e., our final choice). Table 6 shows the results. **Hybrid** always achieves the best accuracy in both datasets, which verifies that our user encoder not only (1) effectively aggregates the embeddings of a user's clicked news articles (i.e., content) by considering their different significance but also (2) leverages mutual complementary information (i.e., collaborative signals), as we claimed in Section 4.2.

We also evaluate the impact of the GNN-enhanced user encoder on the accuracy according to different GNN models. For the GNN

Table 7: News recommendation accuracy for cold-start users: CROWN consistently outperforms *all* competing methods in terms of *all* metrics (The bold font and underline indicate the best and the second best results, respectively).

| | | LibFM | DSSM | NPA | NRMS | NAML | LSTUR | FIM | HieRec | CNE-SUE | DIGAT | GLORY | **CROWN** | Gain |
|---|---|---|---|---|---|---|---|---|---|---|---|---|---|---|
| **MIND** | AUC | 0.5020 | 0.5832 | 0.6022 | 0.6021 | 0.6118 | 0.5908 | 0.6073 | 0.6104 | 0.5985 | 0.5826 | 0.6092 | **0.6352** | +3.82% |
| | MRR | 0.2509 | 0.2810 | 0.3080 | 0.3123 | 0.3139 | 0.3003 | 0.3066 | 0.3128 | 0.2945 | 0.2952 | 0.3081 | **0.3263** | +3.95% |
| | nDCG@5 | 0.2592 | 0.3036 | 0.3351 | 0.3377 | 0.3405 | 0.3238 | 0.3389 | 0.3425 | 0.3275 | 0.3208 | 0.3397 | **0.3595** | +4.96% |
| | nDCG@10 | 0.3209 | 0.3670 | 0.3951 | 0.3976 | 0.4017 | 0.3861 | 0.3959 | 0.3986 | 0.3873 | 0.3804 | 0.3964 | **0.4187** | +4.23% |
| **Adressa** | AUC | 0.5273 | 0.6045 | 0.6186 | 0.6351 | 0.6718 | 0.6787 | 0.6252 | 0.6682 | 0.6854 | 0.6327 | 0.6723 | **0.7236** | +5.57% |
| | MRR | 0.2863 | 0.3181 | 0.3373 | 0.3645 | 0.3872 | 0.3894 | 0.3436 | 0.3823 | 0.4026 | 0.3502 | 0.3922 | **0.4284** | +6.41% |
| | nDCG@5 | 0.2561 | 0.3432 | 0.3529 | 0.3714 | 0.3932 | 0.4025 | 0.3651 | 0.3819 | 0.4153 | 0.3621 | 0.3948 | **0.4469** | +7.61% |
| | nDCG@10 | 0.3358 | 0.3530 | 0.3728 | 0.3935 | 0.4260 | 0.4384 | 0.3885 | 0.4175 | 0.4460 | 0.3973 | 0.4365 | **0.4752** | +6.55% |

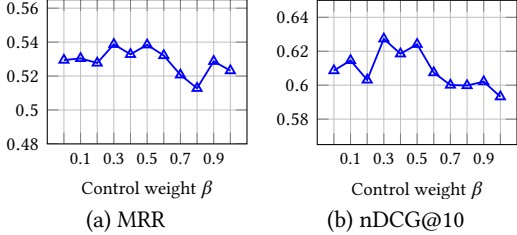

(a) MRR    (b) nDCG@10

Figure 4: The impact of the auxiliary task on the accuracy according to the control weight $\beta$. The auxiliary task with $0.3 \leq \beta \leq 0.5$ is beneficial to improving the accuracy.

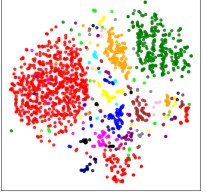 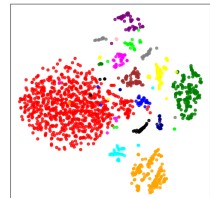

(a) Without the auxiliary task  (b) With the auxiliary task

Figure 5: Visualization of the effect of the auxiliary task of CROWN. News embeddings (a) without and (b) with incorporating the auxiliary task in CROWN.

model, we consider four state-of-the-art GNN models (GCN [11], GraphSAGE [6], GAT [28], and LightGCN [7]). We observed that the GNN-enhanced user encoder is effective in improving the accuracy of CROWN regardless of the GNN models (See Appendix A.5).

## 5.5 EQ4. Auxiliary Task

We evaluate the impact of the auxiliary task (i.e., category prediction) on the accuracy of CROWN according to the control weight $\beta$. We measure the accuracy of CROWN with varying $\beta$ from 0.0 (i.e., not used) to 1.0 (i.e., as the same as the primary task) in step of 0.1. Figure 4 shows the results, where the $x$-axis represents the control weight $\beta$ and the $y$-axis represents the accuracy. The accuracy of CROWN tends to increase until $\beta$ reaches to 0.4 and CROWN achieves the best accuracy at around $0.3 \leq \beta \leq 0.5$. However, the accuracy of CROWN decreases when $\beta$ is larger than 0.5 and CROWN with $\beta \geq 0.6$ shows lower accuracy even than CROWN with $\beta = 0$ (i.e., the auxiliary task is not used). This result verifies that the auxiliary task, i.e., category prediction, provides useful complementary signal to CROWN to be trained for better intent disentanglement. However, too large $\beta$ may cause the model parameters of CROWN to overfit the auxiliary task, i.e., category prediction, rather than the primary task, i.e., click prediction. Figure 5 visually illustrates the effect of the auxiliary task, where the points with the same color indicate the news articles that belong to the same category in **MIND**. Clearly, when the auxiliary task is incorporated in the training process of CROWN, the news embeddings are better learned to distinguish across their categories.

## 5.6 EQ5. Cold Start User Problem

Finally, we evaluate the recommendation accuracy on cold-start users. As cold-start users, we select the users with less than five clicked news articles on two datasets. Then, we measure the recommendation accuracy of all methods for the selected users on the

two datasets. Table 7[3] shows that CROWN achieves the highest accuracy for the cold-start users in all metrics in both datasets. Interestingly, the accuracy degradation in CROWN, compared to the original results shown in Figure 4, is significantly lower than the best and the second-best competitors, i.e., GLORY and CNE-SUE. We analyze the reason why CROWN is superior to existing methods in addressing the cold-start user problem as follows: CROWN is able (1) to precisely comprehend a user's preference via the proposed news encoder even with a small number of clicked news articles and (2) to additionally leverage collaborative signals via the GNN-enhanced hybrid user encoder.

## 6 CONCLUSION

In this paper, we point out three challenges of personalized news recommendation: (C1) comprehension of manifold intents of a news article, (C2) differentiation of users' varying post-read preferences to news articles, and (C3) a cold-start user problem. To tackle these challenges together, we propose a novel approach, named as CROWN that employs (1) the category-guided intent disentanglement for (C1), (2) the consistency-based news representation for (C2), and (3) the GNN-enhanced hybrid user representation for (C3). Furthermore, we integrate the category prediction into the training process of CROWN as an auxiliary task, which provides supplementary supervisory signal to guide the news encoder for better intent disentanglement. Comprehensive experiments on two real-world datasets reveal that (1) (*Accuracy*) CROWN consistently outperforms all competing methods in the news recommendation accuracy and (2) (*Effectiveness*) all proposed strategies are significantly contribute to improving the accuracy of CROWN.

---

[3]We could not include the results of MCCM in this experiment as there is no reported accuracy of MCCM for cold-start users.

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

# A  APPENDIX

In this appendix, we provide the space and time complexity of CROWN (Appendix A.1), the implementation details (Appendix A.2), the experimental results about the impact of the number of intents $K$ on the accuracy of CROWN (Appendix A.3), the qualitative analysis for justifying the category-guided intent disentanglement (Appendix A.4), and the effectiveness of the GNN-enhanced user encoder according to GNN models (Appendix A.5).

## A.1  Complexity Analysis

In this section, we analyze the space and time complexity of CROWN.

**Space Complexity.** CROWN consists of (1) a news encoder, (2) a user encoder, (3) a click predictor, and (4) a category predictor. Storing the parameters of a news encoder requires $4d^2 + 2d^2 + 2d^2 \times K$ for the parameters of MAB, category embedding, and intent disentanglement layers, where $d$ is the embedding dimensionality and $K$ is the number of intents. The parameter size of a user encoder is $d \times |U| + d \times |N| + d^2$ since the space for storing user and news embeddings and for the parameters of a GNN model. The parameter sizes of click and category predictors are $2d$ for dot product of news and user embeddings and $d \times |c|$ for the category prediction layer, respectively, where $|c|$ is the number of categories. Therefore, since $K$, $|c|$ is much smaller than $d$, $|U|$ and $|N|$, the overall space complexity of CROWN is $O((|U|+|N|+d) \cdot d)$ (i.e., linear to the number of users and news articles). Although CROWN requires additional space overhead, especially for the category-guided intent disentanglement layer, the overall space complexity of CROWN is still comparable to that of existing news recommendation methods [4, 16, 21] since the additional overhead is much smaller than the common space overhead (i.e., $O(4d^2) + O(2d^2 \times K) \ll O((|U| \cdot d) + O(|N| \cdot d))$.

**Time Complexity.** The computational overhead of CROWN comes from (1) news representation, (2) user representation, and (3) click and category prediction. The computational overhead of news representation is $O(n_q \cdot n_k + d^2)$ for the MAB and category disentanglement, where $n_q$ and $n_k$ are the number of words in the news title/content. Following [12], we adopt the induced MAB to reduce the computational overhead of MAB. Thus, the time complexity of the news representation in CROWN is $O(m(n_q + n_k) + d^2)$, where $m \ll \min(n_q, n_k)$. The GNN-based update for user representation requires $O(|A| \times d \times k)$, where $|A|$ is the number of non-zero elements in the user-news bipartite graph. Lastly, the click and category prediction require $O(d + d \times |c|)$. Therefore, the overall time complexity of CROWN is $O(|A| \cdot d)$, i.e., linear to the number of users' clicks.

## A.2  Implementation Details

We use PyTorch 1.12.1 to implement CROWN on Ubuntu 20.04 OS. We run all experiments on the machine equipped with an Intel i7-9700k CPU with 64 GB memory and a NVIDIA RTX 2080 Ti GPU, installed with CUDA 11.3 and cuDNN 8.2.1. We set the batch size $b$ as 16 (i.e., 16 positive and 16*4 negative examples) for **MIND** and **Adressa** datasets to fully utilize the GPU memory. We set the number of negative examples $M$ as 4, following [21, 24, 25, 33, 38]. We set the number of disentangled intents $K$ as 3 and the weight factor for the auxiliary task $\beta$ as 0.3. Table 8 shows all

the hyperparameters used in our experiments. For reproducibility, we also have released the code of CROWN and the datasets at https://anonymous.4open.science/r/CROWN-7B75.

**Table 8: Hyperparameters used in our experiments.**

| Hyperparameter | Value |
|---|---|
| Optimizer | Adam |
| Learning rate $\eta$ | 5e-5 |
| Dropout $d$ | 0.2 |
| Batch size $b$ | 16 |
| Early stopping epochs | 5 |
| # of intents $K$ | 3 |
| Weight factor $\beta$ | 0.3 |
| # of negative examples $M$ | 4 |
| Max title length | 32 |
| Max content length | 128 |
| Max history length | 50 |
| Word embedding | 300 |

## A.3  Impact of Number of Disentangled Intents

In this section, we evaluate the impact of the number of disentangled intents $K$ on the model accuracy of CROWN. We measure the recommendation accuracy of CROWN with varying $K$ from 1 to 10. Figure 6 shows the results with four accuracy metrics, where the $x$-axis represents the number of disentangled intents $K$ and the $y$-axis represents the accuracy. Across all metrics, the accuracy of CROWN tends to increase as $K$ increases, and achieve the best accuracy at $K = 3$. On the other hand, when the number of intents $K$ is larger than 4, the model accuracy of CROWN starts to decrease. These results imply that (1) the intent disentanglement could be beneficial to enhancing the news representation until a specific limit but (2) too many intents might have adverse effect on the news representation.

## A.4  Justification of the category-guided intent disentanglement

In this section, we provide empirical evidence to justify the category-guided intent disentanglement. Specifically, we (1) select news articles across four categories (Politics, Sports, Weather, and Health), (2) represent each news article into $k$ disentangled intent embeddings, and (3) compute the relative importance score of each intent embedding by using an attention mechanism [27]. As illustrated in Figure 7, news articles within the same category (i.e., highlighted as the same color) tend to show similar intent distributions, even if their news contents differ, whereas news articles from different categories exhibit different intent distributions. These results support our claim that the intent of news articles is closely related to their category. Interestingly, the news articles in the category 'Sports' and 'Weather' show similar intent distributions. More specifically, their most important intents are the same (i.e., the first intent, the darkest color). We analyze that this result arises because the most important intent of both news categories (i.e., 'Sports' and 'Weather') is to provide information to readers. Therefore, through

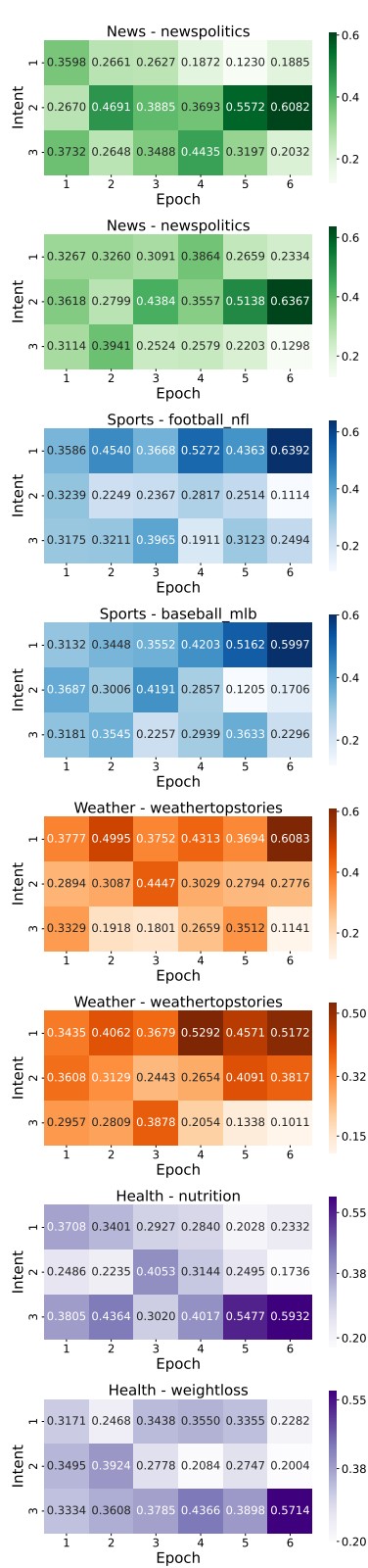

Figure 7: Different intent distributions of news articles according to their categories (Politics, Sports, Weather, and Health).

Table 9: The effectiveness of the GNN-enhanced user encoder on the accuracy of CROWN according to different GNN models on the MIND dataset.

| | | Baseline | GCN | Gain | GraphSAGE | Gain |
|---|---|---|---|---|---|---|
| **MIND** | AUC | 0.6720 | 0.6821 | +1.50% | 0.6823 | +1.53% |
| | MRR | 0.3194 | 0.3350 | +4.88% | 0.3354 | +5.01% |
| | nDCG@5 | 0.3550 | 0.3692 | +4.00% | 0.3697 | +4.14% |
| | nDCG@10 | 0.4174 | 0.4304 | +3.11% | 0.4293 | +2.85% |

| | | Baseline | LightGCN | Gain | GAT | Gain |
|---|---|---|---|---|---|---|
| **MIND** | AUC | 0.6720 | 0.6846 | +1.88% | 0.6857 | +2.04% |
| | MRR | 0.3194 | 0.3380 | +5.82% | 0.3405 | +6.61% |
| | nDCG@5 | 0.3550 | 0.3745 | +5.49% | 0.3783 | +6.56% |
| | nDCG@10 | 0.4174 | 0.4364 | +4.55% | 0.4384 | +5.03% |

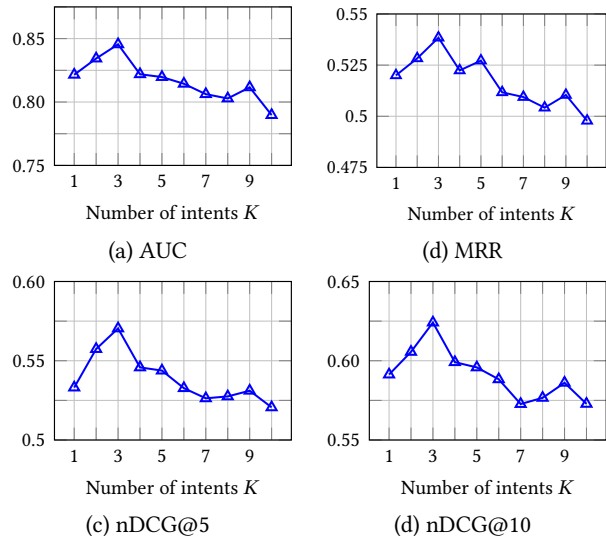

(a) AUC          (d) MRR

(c) nDCG@5          (d) nDCG@10

Figure 6: The impact of the number of disentangled intents $K$ on the model accuracy of CROWN.

the category-guided intent disentanglement, CROWN can understand the intents of a news articles more precisely, thereby leading to better news representation.

## A.5  Effectiveness of Our GNN-enhanced User Encoder

In this section, we evaluate the effectiveness of our GNN-enhanced user encoder by examining whether the GNN modules improve the accuracy of CROWN. We measure the recommendation accuracy of four CROWNs with different GNN models (i.e., GCN [11], GraphSAGE [6], GAT [28], and LightGCN [7]) and compare them with the baseline version of CROWN (i.e., CROWN with a simple average user encoder, rather than our GNN-enhanced user encoder). As shown in Table 9, the GNN-enhanced user encoder is consistently beneficial to improving the news recommendation

accuracy of CROWN, regardless of the GNN models. More specifically, CROWN with the GNN-enhanced user encoder improves its baseline version by 4.88% (GCN), 5.01% (GraphSAGE), 5.82% (LightGCN), and 6.61% (GAT) respectively, in terms of MRR. Based on these results, we choose GAT as the final GNN model for the GNN-enhanced user encoder in CROWN. It is worth noting that the recommendation accuracy of CROWN could be further improved by leveraging more recent state-of-the-art GNN models or future GNN models.

