# OpenReview forum: "CROWN: A Novel Approach to Comprehending Users' Preferences for Accurate Personalized News Recommendation"
_ACM.org/TheWebConf/2025/Conference — WWW 2025 Oral_

### Official Review · Reviewer_nuqt · 2024-11-21

**Novelty:** 5
**Technical Quality:** 5

**Review:**

Summary:

This work addresses three main challenges in personalized news recommendations: understanding the multiple intents of a news article, recognizing the different preferences users have after reading, and handling the cold-start problem with new users. To tackle these issues, the authors introduce a new approach called CROWN, which combines category-guided intent disentanglement, consistency-based news representation, and GNN-enhanced hybrid user representation. Through extensive experiments on two real-world datasets, the authors answer five research questions, demonstrating CROWN's advantages over other benchmark models.

Technical Quality:

The paper shows strong technical quality, with a clear and structured approach to addressing challenges in news recommendation systems. It introduces well-designed methods, like intent disentanglement and category-guided news representation, backed by detailed mathematical explanations and clear visuals. For the news encoder, the authors present hypotheses with statistical evidence and experiments to support them, which strengthens the foundation for their methods. The “Problem Definition” section is well-organized, outlining two main challenges and the solutions proposed to address them. The CROWN framework is clearly structured, with different modules thoroughly explained, and the diagrams further clarify the concepts. The experiments are extensive, using two real-world datasets to validate the approach, and the results show that this framework outperforms all benchmark models.


Clarity:

The paper is well-organized and clearly written. Key terms and background information are explained clearly in the Introduction, making it easier to follow. The diagrams and tables are well-labeled and explained, and the modular breakdown of the model helps make complex ideas more understandable.


Originality:

This work presents a unique approach to improving recommendation accuracy by using category-guided intent disentanglement. The focus on maintaining consistency between a user’s click history and the recommended news is a fresh and valuable contribution to news recommendation models. Additionally, the paper's method of separating the intents within a news article for better representation is an innovative feature.


Significance:

This model has the potential to make a substantial impact on personalized news recommendations. By focusing on consistency and intent-based recommendations, the paper addresses common issues in news recommendation accuracy, which is crucial for applications in content delivery, particularly in news platforms.

Pros:

•	The related works are well-explained, with a clear comparison of the proposed method against existing approaches, supported by a table.

•	The main challenges and the authors' contributions are outlined clearly, with solutions effectively introduced in the Introduction.

•	The methodology is well-structured, with distinct modules for encoding news and user data.

•	The approach is technically rigorous and includes clear and understandable mathematical formulations.

•	Baselines are well-categorized, and all five research questions are thoroughly addressed.

•	An experiment in the Appendix examines the impact of the number of disentangled intents on performance.

•	Ablation studies are provided to verify the effectiveness of the proposed strategies in the news encoder.

•	The experiments demonstrate how effectively CROWN addresses the cold-start problem.

•	The Appendix includes an analysis of computational efficiency, covering both time and space complexity.

•	Through experiments and statistical analysis, the authors confirm their hypotheses, enabling them to choose effective methods for the News and User Encoders.

•	The user encoder's effectiveness in enhancing recommendation accuracy is also demonstrated through experiments.


cons

•	The paper uses complex terminology and dense sections, which may be difficult for some readers to follow.

•	There is a lack of clear explanations or examples showing how intent disentanglement directly impacts user experience.

•	It would improve clarity to explain the CROWN methodology (four modules) in detail only in the methodology section, with the Introduction focusing on main contributions without going into specifics.

•	The paper offers limited exploration of potential limitations in the proposed model.

**Questions:**

• Could you provide a clear definition of "intent disentanglement" and "disentangled embeddings" in the Introduction? These terms currently seem unrelated to neural networks before section 4.1.

• Why were these specific benchmarks chosen for comparison? There are many other methods for news recommendations on the same datasets. Are there other benchmarks that might show better performance than CROWN?

• Can you clarify how intent disentanglement improves user experience or recommendation quality, especially compared to traditional methods?

• What are the specific limitations of this model, and in what situations might it perform less effectively than other news recommendation models?

**Ethics Review Description:**

The submission does not meet the double-blind review requirements specified in ACM Web's Call for Papers (CFP). While the paper's content appears as a preprint on arXiv (2310.09401v3), the issue lies not with the preprint's existence but with its non-anonymized format. According to ACM Web's submission policy, authors are permitted to have preprints available on platforms such as arXiv or SSRN. However, these preprints must also be anonymized.

**Ethics Review Flag:**

Yes

**Reviewer Confidence:**

3: The reviewer is confident but not certain that the evaluation is correct

**Scope:**

4: The work is relevant to the Web and to the track, and is of broad interest to the community

---

### Official Review · Reviewer_WxPx · 2024-11-25

**Novelty:** 5
**Technical Quality:** 5

**Review:**

This paper aims to solve two challenges of news recommender systems that have been underexplored, in combination with the traditional challenge of the cold start user problem. The first new challenge is considering the different intentions underlying the news articles in the recommendation, the second - how to differentiate user interest from their click history. For example, when the user clicks the heading of a news article which appears interesting and reads it, contrasted with the case when the user clicks the title, but immediately closes it and moves on. In both cases, the clicked article is stored in the history and will be used to model the user's interest, but the second case should indicate a lack of interest in the use, not the opposite.

I find these two new ideas intriguing, but I was not convinced that it was worthwhile exploring either of them. There was no example showing on an intuitive level how considering the intention of the news article (inform, persuade, ?) can improve the recommendation. One can argue that the intent of an article is often hard to identify, even for humans, and it is not clear what decision should be made based on the detected intent - to recommend or not.  Similarly, for the second challenge, opening and closing an article may not necessarily indicate a lack of interest. The user may be interested but have a conflicting event at the time; if the article is long, for example, the reader may want to read it later but forget and never get back to it. So, opening and closing should not necessarily indicate that the user has no interest in the article.

The authors propose a framework for personalized news recommendation that addresses the three challenges and evaluates it on real-world datasets with respect to accuracy by comparing it with existing state-of-the-art methods.

I find the paper ambitious and interesting. It is excellently presented, with nice illustrations and good organization that defines the main problems and approaches intuitively in the introduction before presenting the literature review, the formal definition of the problem and the proposed framework and its evaluation. The latter parts of the paper are highly technical, and I did not delve deep into trying to understand the details, but it seems that the notations are appropriately used and the math is correct.

The evaluation results look impressive, but I don’t understand how the mathematics described in section 4 was applied to produce these results.  A high-level description of the system design would have been helpful, though I realize it is hard to fit in the limited space. The authors have provided a link to the code and the datasets for reproducibility, which is praiseworthy.

**Questions:**

I had the following questions while reading the paper:
•	What is the range of possible intents?  What does "Intent 1, 2 or 3" mean in the figures? An explanation and some examples would be useful.  Categories Sports, Political, Weather (as in Figure 2), and Health (In Figure 8 in Appendix 4.2), are shown, but the range of intents listed in the paper is very narrow: informing and persuading.  What about other possible categories: Business, Environment, Education, Culture, Books,  Movies, Lifestyle, Cars? Would there be different intents in these categories? Who defines these intentions and who provides the ground data for training the encoders?
•	Does the approach scale up for more categories and possible intents within each category? Showing the possible range of intents in a wider range of categories can help evaluate the significance of the approach.
•	Two real-world datasets were used in the evaluation: MIND-small and Adressa-1week. It would have been nice to provide some more information about those datasets, since readers may not be familiar with them. Do they contain annotations about the categories and, more importantly, about the intents in each new article?

**Ethics Review Description:**

The paper has been published in identical form and not anonymized on ArXiv 2310.09401v3 (https://arxiv.org/pdf/2310.09401). This violates the Originality and Concurrent Submission Policy in the CFP. It is easy very easy to find the non-anonymized preprint paper by searching for the title.

**Ethics Review Flag:**

Yes

**Reviewer Confidence:**

2: The reviewer is willing to defend the evaluation, but it is likely that the reviewer did not understand parts of the paper

**Scope:**

4: The work is relevant to the Web and to the track, and is of broad interest to the community

---

### Official Review · Reviewer_US8C · 2024-11-26

**Novelty:** 4
**Technical Quality:** 5

**Review:**

The paper introduces CROWN, a framework designed to tackle three core challenges in personalized news recommendation:

1. Comprehending manifold intents in news articles.
2. Differentiating users' post-read preferences.
3. Addressing the cold-start user problem.

The proposed framework integrates three novel strategies:

- Category-guided intent disentanglement for improved news representation.
- Consistency-based news representation to reflect users’ reading preferences.
- GNN-enhanced hybrid user representation for cold-start scenarios.

Experiments conducted on two real-world datasets (MIND and Adressa) demonstrate that CROWN consistently outperforms twelve state-of-the-art models across multiple metrics, including AUC, MRR, and nDCG.

Pros:

- Well-written and easy-to-follow style.
- Effective portrayal of user attitudes towards clicked news without relying on other user feedback.
- Innovative Multi-Module Design: Combines category-guided intent disentanglement, consistency-based representation, and graph-based hybrid models to address distinct recommendation challenges.
- Extensive Experiments: Conducted comparisons with multiple baselines on two datasets and performed ablation studies to evaluate the effectiveness of each module.

Cons:

- Can category always represent intent distribution? What if some news articles within the same category have different intents? For example, a sports news article aimed at gaining support for a sports star may have a higher intent similarity with a political news article aimed at persuading people to support certain political leaders, than with a sports news article reporting the Olympic results from an official source?
- Higher growth in MRR and NDCG on the Adressa dataset for warm users compared to cold-start users, suggesting that the improvement for warm users is greater than that for cold-start users. It seems that the GNN's primary benefit is enhancing overall user representation effectiveness, rather than benefiting cold-start users more.
- Increased demand for computational resources due to the use of multiple intent embeddings and attention integration.

**Questions:**

Refers to the Cons.

**Reviewer Confidence:**

3: The reviewer is confident but not certain that the evaluation is correct

**Scope:**

4: The work is relevant to the Web and to the track, and is of broad interest to the community

---

### Official Review · Reviewer_evDx · 2024-11-29

**Novelty:** 6
**Technical Quality:** 5

**Review:**

This manuscript tackles three challenges in news recommendation systems: comprehending diverse news intents, differentiating reading preferences, and the cold-start problem. The authors propose a novel model called CROWN, which overcomes these challenges through category-guided disentanglement, consistency-based representation, and GNN-enhanced user modeling. Additionally, CROWN incorporates category prediction as an auxiliary task during training to strengthen intent disentanglement. Experiments demonstrate its superior accuracy compared to state-of-the-art methods on two datasets.

Pros:
1) Consistency-based Representation: The concept of consistency-based news representation is innovative and addresses the varying post-read preferences of users, providing a more nuanced approach to news article recommendations.

2) Integration of Auxiliary Task: The incorporation of category prediction as an auxiliary task is a clever strategy that enhances intent disentanglement.

3) Robustness Across Datasets: The paper demonstrates the effectiveness of CROWN across two real-world datasets, showcasing its robustness and generalizability.

4) Clear Structure and Comprehensive Explanation: The manuscript is well-organized, with a clear presentation of the methodology and results. The explanations are easy to follow, which enhances the readability and understanding of the paper.

Cons:
1) Insufficient Innovation in GNN Application and Cold-Start Challenge: The manuscript employs a Graph Neural Network (GNN) to enhance the user-news bipartite graph, followed by an attention-based fusion, which, while effective, is not a novel approach. As discussed in “Graph Neural Networks in Recommender Systems: A Survey”, this method of user-item collaborative filtering is well-documented, diminishing the perceived innovation of the approach. Additionally, the paper could provide a more detailed exploration of the cold-start problem in the context of news recommendation, explaining the specific challenges it poses compared to other domains.

2) Insufficient Experiments for the Post-Read Preferences (C2) Challenge: Regarding the C2 issue, since the experiments in the manuscript primarily rely on news title embeddings for recommendations, and the selection of positive samples and evaluation metrics still depend on click history, it is unclear whether the CROWN model might recommend news articles to users when the title does not align with the actual content (i.e., title deception). The manuscript does not thoroughly address this potential issue, which prevents validation of CROWN’s effectiveness in addressing the C2 challenge (i.e., differentiating news articles with different post-read preferences). It is suggested that the authors consider and discuss the impact of title deception on the model’s recommendation performance and whether further improvements are needed to mitigate such issues.

**Questions:**

1) Could the authors clarify whether the category prediction task in the auxiliary task is limited to the main category, or does it also include the subcategory information? Additionally, could the authors explain the necessity of incorporating subcategories into the model? Specifically, what advantages does including subcategory information bring to the performance of the news encoder?

2) The manuscript states that the best performance is achieved when the number of intents K is set to 3. However, intuitively, applying the same fixed number of intents to all news articles does not seem entirely reasonable. Could the authors elaborate on why this design choice of a fixed number of intents was made? Furthermore, is relying on a fixed number of intents the optimal strategy, or might a dynamic approach that determines the number of intents for each news article individually be more appropriate?

**Reviewer Confidence:**

4: The reviewer is certain that the evaluation is correct and very familiar with the relevant literature

**Scope:**

3: The work is somewhat relevant to the Web and to the track, and is of narrow interest to a sub-community

---

### Official Review · Reviewer_FyTJ · 2024-12-03

**Novelty:** 5
**Technical Quality:** 7

**Review:**

### Summary

The paper presents a personalized news recommendation framework designed to enhance user experience through the following key modules:

1. News Encoder: This module employs consistency loss to train representations of news articles. It assigns positive relations to news within the same category while pushing apart news from different categories, based on the assumption that users are likely to prefer news within a similar category.

2. User Encoder: This model learns user representations from a graph that captures relationships between all users and all news articles. It effectively addresses the cold-start problem by generating meaningful representations for users with limited reading history.

### Strength

The paper is well-written, with clear and concise content.
The proposed framework effectively models personalized news recommendations, leveraging contrastive learning for content categorization and graph-based representations for user personalization.

### Weakness

None

**Questions:**

1. The performance gain appears to be significant when the dataset includes diverse categories, enabling more fine-grained clustering of news articles. Have you considered creating sub-categories for the news articles to further enhance performance? If so, how might this be implemented?

2. Instead of focusing on consistency loss, have you explored whether contrastive learning could further improve performance? It would be interesting to see an analysis or experiment in this direction.

3. You may want to further compare it with other recent baselines
e.g., News Recommendation with Category Description by a Large Language Model., Yada and Yamana., 2024

**Reviewer Confidence:**

4: The reviewer is certain that the evaluation is correct and very familiar with the relevant literature

**Scope:**

4: The work is relevant to the Web and to the track, and is of broad interest to the community